# Questionnaire and Portable Sleep Test Screening of Sleep Disordered Breathing in Acute Stroke and TIA

**DOI:** 10.3390/jcm10163568

**Published:** 2021-08-13

**Authors:** Benjamin K. Petrie, Tudor Sturzoiu, Julie Shulman, Saleh Abbas, Hesham Masoud, Jose Rafael Romero, Tatiana Filina, Thanh N. Nguyen, Helena Lau, Judith Clark, Sanford Auerbach, Yelena G. Pyatkevich, Hugo J. Aparicio

**Affiliations:** 1Neurology Department, Medical Campus, Boston University School of Medicine, Boston, MA 02118, USA; bpetrie@bu.edu (B.K.P.); Tudor.Sturzoiu@jefferson.edu (T.S.); shulman@bu.edu (J.S.); sa3817@cumc.columbia.edu (S.A.); joromero@bu.edu (J.R.R.); Thanh.Nguyen@bmc.org (T.N.N.); sauerbac@bu.edu (S.A.); ygorfinkel@gmail.com (Y.G.P.); 2Department of Internal Medicine, Downtown Campus, McGaw Medical Center of Northwestern University, Chicago, IL 60611, USA; 3Boston Medical Center, Boston, MA 02118, USA; hlau@bu.edu (H.L.); judith.clark@bmc.org (J.C.); 4Department of Neurology, Syracuse Campus, SUNY Upstate Medical University, Syracuse, NY 13210, USA; hesham.masoud@gmail.com; 5Boston Children’s Hospital at Waltham, Waltham, MA 02453, USA; tfilina@bu.edu

**Keywords:** cerebrovascular disorders, stroke, sleep apnea, screening

## Abstract

Sleep disordered breathing (SDB) is highly prevalent, but frequently unrecognized among stroke patients. Polysomnography (PSG) is difficult to perform soon after a stroke. We evaluated the use of screening questionnaires and portable sleep testing (PST) for patients with acute stroke, subarachnoid hemorrhage, or transient ischemic attack to expedite SDB diagnosis and management. We performed a single-center retrospective analysis of a quality improvement study on SDB screening of consecutive daytime, weekday, adult admissions to a stroke unit. We excluded patients who were unable to communicate and lacked available family members. Patients were screened with the Epworth Sleepiness Scale, Berlin Questionnaire, and STOP-BANG Questionnaire and underwent overnight PST and/or outpatient PSG. The 4-item STOP Questionnaire was derived from STOP-BANG for a secondary analysis. We compared the sensitivity and specificity of the questionnaires for the diagnosis of at least mild SDB (apnea hypopnea index (AHI) ≥5) on PST and correlated AHI measurements between PST and PSG using the Spearman correlation. Out of sixty-eight patients included in the study, 54 (80%) were diagnosed with SDB. Only one (1.5%) had a previous SDB diagnosis. Thirty-three patients completed all questionnaires and a PST. The STOP-BANG questionnaire had the highest sensitivity for at least mild SDB (0.81, 95% CI (confidence interval): 0.65–0.92) but a low specificity (0.33, 95% CI 0.10, 0.65). The discrimination of all questionnaires was overall poor (C statistic range 0.502–0.640). There was a strong correlation (r = 0.71) between the AHI results estimated using PST and outpatient PSG among 28 patients. The 4-item STOP Questionnaire was the easiest to administer and had a comparable or better sensitivity than the other questionnaires. Inpatient PSTs were useful for screening in the acute setting to facilitate an early diagnosis of SDB and to establish further outpatient evaluations with sleep medicine.

## 1. Introduction

Sleep disordered breathing (SDB) is an upper airway dysfunction that results in the transient cessation of breathing during sleep and is diagnosed with polysomnography (PSG). While studies estimating the prevalence of SDB have been variable, the prevalence of obstructive sleep apnea (OSA, the most common form of SDB) ranged from 9–38% in a large systematic review. Prevalence of SDB increases with age, while other important risk factors include obesity and male sex [1].

Prevalence of SDB is even higher among acute stroke patients, with nearly three-fourths of patients having comorbid SDB [2]. Stroke patients diagnosed with SDB have a poorer functional recovery and are more likely to die or have a recurrent stroke, especially among nonwhite racial and ethnic populations [3,4]. Underscoring the importance of recognizing SDB in acute stroke and the potential for continuous positive airway pressure (CPAP) treatment to improve stroke outcomes, American Heart Association/American Stroke Association guidelines recommend considering sleep studies in patients with acute stroke and transient ischemic attack (TIA) [5,6]. However, screening is rare after stroke, and outpatient polysomnography (PSG) testing may take weeks or months to perform [7,8].

We studied the use of three common screening questionnaires (Epworth Sleepiness Scale (ESS), the Berlin Questionnaire (BQ), and STOP-BANG (SB)) and an overnight portable sleep test (PST) device to diagnose SDB among inpatients with acute stroke, subarachnoid hemorrhage, and TIA at our medical center. The STOP Questionnaire (STOP), a four-question subset of SB, was also evaluated. We sought to validate these tools for an expedited diagnosis of SDB in our diverse stroke patient population and to facilitate an outpatient evaluation with sleep medicine and/or a polysomnogram.

## 2. Materials and Methods

### 2.1. Patient Sample

Data were collected retrospectively from a quality improvement study initiated at an urban academic medical center to improve the screening of acute stroke inpatients for SDB, performed between October 2014 and October 2015. During the timeframe of the study, 344 patients were admitted to the acute stroke unit. From these admissions, adult daytime, weekday admissions with a diagnosis of ischemic stroke, hemorrhagic stroke, subarachnoid hemorrhage, or TIA were approached for SDB screening. The Boston University Medical Center Institutional Review Board approved this study.

### 2.2. Inclusion and Exclusion Criteria

Adult inpatients with acute stroke, subarachnoid hemorrhage, or TIA were included. Three patient groups were studied: Group 1, consisting of patients who completed all three questionnaires (ESS, BQ, and SB) and had an inpatient PST; Group 2, consisting of patients who completed both an inpatient PST and an outpatient PSG; and Group 3, any patients who completed the SB questionnaire and an inpatient PST.

Patients without an acute cerebrovascular event (ischemic stroke, hemorrhagic stroke, subarachnoid hemorrhage, or TIA) were excluded. We excluded patients admitted to the neurocritical care unit. Patients who did not complete either PST (Groups 1 and 3) or PSG (Group 2) were excluded. We also did not screen for SDB among patients who were unable to communicate and had no available family members to provide responses for the questionnaires.

### 2.3. Questionnaires and Sleep Testing

Patients were administered ESS, BQ, and SB questionnaires (available in the Appendix A) on admission and then underwent inpatient overnight PST and/or outpatient PSG. The STOP questionnaire, which includes four yes/no questions related to SDB (snoring, tiredness, observed apnea, and high blood pressure), is a component of the STOP-BANG questionnaire, which additionally collects data on body mass index (BMI), age, neck circumference, and male sex/gender. Per standard SDB cutoffs using these screening tools, ESS ≥ 10, BQ ≥ 2, SB ≥ 3, and STOP ≥ 2 scores were considered suggestive for SDB [9,10,11,12].

During admission, a ResMed ApneaLink Air (ResMed, Poway, CA, USA) type III home sleep apnea testing device was used, which consists of five channels to measure respiratory effort, respiratory flow, snoring, pulse, and oxygen saturation. The device consists of a respiratory effort sensor belt, a nasal pressure cannula sensor, and a finger probe for oximetry and pulse measurement. Recordings were scored by certified sleep technicians following American Academy of Sleep Medicine guidelines, and a sleep medicine physician reviewed and interpreted data from the ResMed AirView online platform. The criteria for sleep apnea included (1) baseline oxygen saturation >90%, (2) artifact-free recording of arterial oxygen saturation ≥1 h, (3) <10% overnight recording time with arterial oxygen saturation of ≤80%, and (4) oxygen desaturation index (ODI) of ≥3% at ≥12 events per hour.

A digital polygraph (Nihon Kohden Co., Irvine, CA, USA) was used for PSG. Certified technicians scored sleep staging and arousals, and a Pro-Tech nasal pressure transducer (Natus Medical Inc., Middleton, WI, USA) recorded airflow. Respiratory impedance plethysmography was used during the diagnostic phase of the PSG. The definition for apnea included cessation of airflow ≥10 s, while the definition for hypopneas included 30% declines in airflow lasting ≥10 s with at least 3% oxyhemoglobin desaturation [13].

For the definition of at least mild SDB and OSA, an apnea-hypopnea index (AHI) ≥5 per hour was used as a standard and inclusive cutoff [14,15]. Central sleep apnea (CSA) was defined as an observation of a central apnea index of ≥5 per hour and when ≥50% of the AHI were purely central. We defined Cheyne-Stokes breathing as ≥3 consecutive cycles of cyclical crescendo/decrescendo change in the amplitude of breathing, with ≥5 central apneas or hypopneas per hour or with a duration of ≥10 min. Patients considered to have SDB included individuals with only OSA, only CSA, and both OSA and CSA.

### 2.4. Statistical Analysis

Retrospective analyses of these data were performed. The baseline characteristics within these groups were described and included: age, BMI, neck circumference, sex, race (white or Black), ethnicity (Hispanic or Asian), hypertension, diabetes, and prior diagnosis of SDB. The reason for acute admission was categorized into ischemic stroke, intracerebral hemorrhage, subarachnoid hemorrhage, or TIA. The proportion of patients with at least mild SDB was calculated using the threshold of AHI ≥5 per hour, and that of moderate-to-severe SDB using AHI ≥15 per hour. Patients with SDB were further subcategorized into having a diagnosis of obstructive sleep apnea, central sleep apnea, or a combination of obstructive sleep apnea and central sleep apnea, based on either PST study results (Groups 1 and 3) or PSG (Group 2). The distributions of the above variables were described within each group, allowing for a between-group comparison.

The two primary analyses for this study were (1) the assessment of the validity of the screening questionnaires for the diagnosis of mild SDB, describing the sensitivity and specificity and 95% confidence intervals (CI) of these tools, as well as the C statistic to assess for discrimination ability (Groups 1 and 3), and (2) the use of the Spearman correlation to compare individual patient AHI measurements between PST and PSG (Group 2). We performed secondary analyses using the 4-item STOP questionnaire, with cutoffs at ≥1, ≥2, and ≥3 points. Receiver operating characteristic (ROC) curves were constructed to display the diagnostic ability of all questionnaires and alternative STOP cutoffs for at least mild SDB (AHI ≥5). The analysis was performed using R version 3.5.1 (R Foundation for Statistical Computing, Vienna, Austria).

## 3. Results

During the screening program, 68 patients completed at least one of the questionnaires, PST, or PSG. Thirty-three patients completed all three questionnaires and had inpatient PST performed (Group 1); 28 completed both PST and PSG (Group 2). In addition, 49 patients overall completed both the SB questionnaire and a PST (Group 3). On average, patients underwent PST 1.8 (SD ± 1.8) days after admission and had PSG performed 64 (SD ± 36) days after admission.

Table 1 displays the patient characteristics, questionnaire results, and sleep study results in each of the three groups. Across groups, patients were middle-aged, more than half were of the male sex (50–61%), and approximately two-thirds were Black patients. In Group 1, the BMI was on average 29.7 (SD ± 5.8), and the neck circumference was on average 38.4 (SD ± 5.0) cm. Of the 68 patients who underwent PST and/or PSG, 54 (80%) patients were diagnosed with at least mild SDB (AHI ≥5). Nearly half of the patients in each group had at least moderate SDB (42–57%). Only one patient (1.5%) had a previous diagnosis of SDB (OSA).

In the primary analysis, the SB questionnaire had the highest sensitivity for at least mild SDB (0.81, 95% CI: 0.65–0.92) compared to the BQ and ESS (Table 2). However, SB had a low specificity (0.33, 95% CI 0.10–0.65). None of the questionnaires had a good discrimination capacity to distinguish between a positive or negative diagnosis of SDB on PST (Figure 1). The C-statistics for SB, ESS, and BQ were 0.572, 0.502, and 0.640, respectively.

In the secondary analysis, positive responses to at least 1 (SO ≥1) or at least 2 (SO ≥ 2) questions on the STOP Questionnaire had sensitivities for SDB (0.95 and 0.68, respectively) that were comparable to SB and BQ (Table 3). Like the other questionnaires in the primary analysis, the STOP Questionnaire with cutoffs of ≥1, ≥2, and ≥3 points showed poor discrimination (0.515, 0.588, and 0.565, respectively).

The Spearman coefficient between the PST and PSG AHI results was 0.71. Of the eight patients diagnosed with central sleep apnea on PST, six patients (75%) were found to have central sleep apnea on PSG during outpatient follow-up. Three new cases of central sleep apnea were found on PSG where this condition was not diagnosed initially on PST. Of the nine patients with central sleep apnea on PSG, four were diagnosed with Cheyne-Stokes breathing.

## 4. Discussion

We observed a high prevalence of at least mild SDB (80%) in our diverse population of acute stroke, subarachnoid hemorrhage and TIA patients, implementing the use of a home sleep apnea testing device for early diagnosis during admission. Of the screening questionnaires, STOP-BANG had the highest sensitivity, but had a low specificity for SDB, and none of the questionnaires (BQ, ESS, SB, or STOP) were able to discriminate effectively between patients with or without SDB. In our experience, the 4-item STOP Questionnaire was the easiest and quickest to administer upon admission, while also having a good sensitivity so that fewer cases of SDB were missed. The PST and PSG AHI measurements were strongly correlated, despite PSG being performed on average over two months later.

Our findings support prior conclusions that SDB and stroke are highly comorbid, although under-screening and under-diagnosis are common, especially within nonwhite racial and ethnic populations [16,17,18]. Only one patient out of 68 in our cohort had a previous diagnosis of SDB. Compared to ESS (eight items), BQ (10 items), and SB (eight items), the 4-item STOP Questionnaire is a simple, brief, and sensitive screening method that can be administered to most acute stroke patients within the context of an admission (Figure 2). Although SB was predicted a priori to be the best performing questionnaire for the purposes of the QI (quality improvement) project, the requirement of measuring the neck circumference using a tape measure made the administration more cumbersome and less ideal than the simplified STOP Questionnaire [19]. All questionnaires, at all cutoffs, had a suboptimal sensitivity and/or specificity for use as screening measures to diagnose SDB. Clinicians should have a low index of suspicion for further testing even with negative results on the questionnaire. Patients should undergo formal sleep study testing to accurately diagnose OSA and CSA in this population. Of note, while the focus of our study was on SDB overall, SB has been more thoroughly studied in OSA compared to CSA.

While no inpatient test has sufficient sensitivity to obviate the need for a PSG in the acute stroke population, PST has been shown in our study and in others to be highly consistent with PSG and can expedite the diagnosis of SDB [20]. We outline our proposed algorithm for SDB screening in acute stroke in Figure 2. Patients should be initially screened with the STOP Questionnaire, with cutoffs of ≥1 or ≥2. Patients with a negative STOP (score of 0) are less likely to have SDB, although nonurgent sleep medicine referral and PSG are recommended. Patients with a PST related to isolated, moderate or severe obstructive sleep apnea are likely to have SDB and may proceed to CPAP autotitration as outpatients. Since CPAP autotitration is inappropriate for those with CSA, we referred patients with PST related to CSA to sleep medicine. If PST is unavailable, patients who screen positive via the STOP Questionnaire may also undergo a split-night polysomnogram with referral to sleep medicine for diagnosis and CPAP titration if the AHI is over 15–20/events per hour. Clinical trials, including the Sleep SMART (Sleep for Stroke Management and Recovery Trial) trial, are underway to evaluate the outcomes of the initiation of CPAP in an acute stroke setting [21].

One of the limitations of our study is its small sample size. The results of the analyses should be considered as hypothesis-generating. Follow-up studies should include larger sample sizes or a more longitudinal evaluation for SDB in the acute stroke population. Additionally, as discussed above, none of the questionnaires were by themselves sufficiently sensitive to rule out SDB, which is why we prefer either inpatient PST or outpatient PSG in the follow-up for most stroke patients. Nonetheless, the STOP Questionnaire was simple to use and simple for screening and for prompting further follow-ups. While AHI ≥5 is a cutoff for mild SDB, only ODI 4% ≥12 (equivalent to approximately AHI ≥18) has been associated with an increased risk of stroke [15,22]. We opted for an inclusive screening strategy, although it is worth noting that the adverse effects of SDB on stroke outcomes are likely concentrated in higher AHI groups.

New questionnaires for the screening of SDB continue to be developed. One promising questionnaire is NoSAS, which was developed after our QI study was completed. NoSAS should be compared to the STOP Questionnaire in future studies within stroke populations to see if there is an improvement in the diagnosis of SDB [23]. The development of new questionnaires, advancements in technology facilitating overnight and inpatient sleep testing, and results from clinical trials such as Sleep SMART will inform the diagnosis and treatment of this highly prevalent disorder among vulnerable stroke patients.

In conclusion, there was a high prevalence of SDB in our acute stroke population, and the use of questionnaires and PST were helpful for screening patients during admission. The findings contribute to the literature on sleep apnea screening in acute stroke. While questionnaires have a limited use for making a formal diagnosis, PST is a tool that can be used to make informative diagnoses that can result in further evaluation with sleep medicine to treat this important modifiable risk factor.

## Figures and Tables

**Figure 1 jcm-10-03568-f001:**
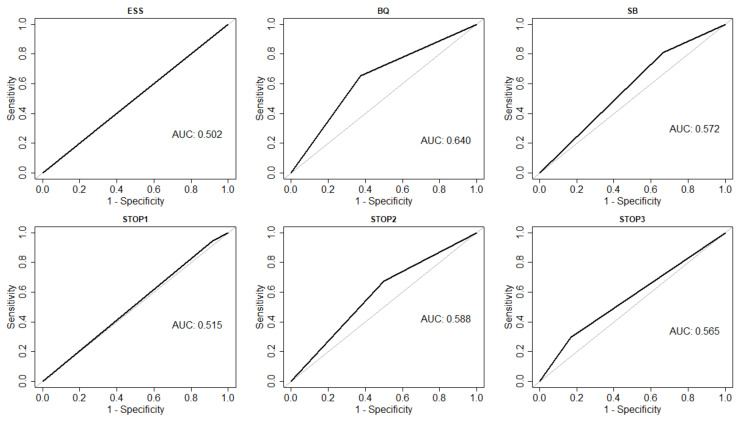
Questionnaire ROC Curve Analysis. Receiver operating characteristic curve for questionnaires constructed from the data presented in Table 2 and Table 3. Abbreviations: ROC, receiver operating characteristic; AUC, area under the curve; BQ, Berlin Questionnaire; ESS, Epworth Sleepiness Scale; SB, STOP-BANG Questionnaire; STOP1, STOP Questionnaire using a cutoff of one; STOP2, STOP Questionnaire using a cutoff of two; STOP3, STOP Questionnaire using a cutoff of three.

**Figure 2 jcm-10-03568-f002:**
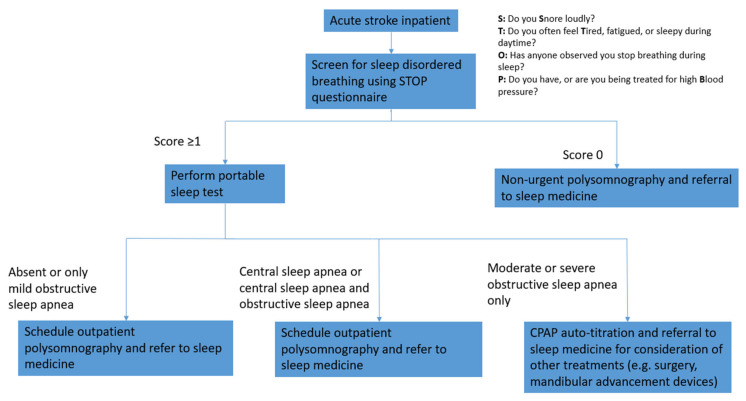
Sleep Disordered Breathing Screening Algorithm. Our proposed algorithm for screening patients for sleep disordered breathing after an acute stroke.

**Table 1 jcm-10-03568-t001:** Patient Characteristics, Questionnaires, and Sleep Study Results, by Group. Table comparing acute stroke, subarachnoid hemorrhage, and transient ischemic attack patient clinical characteristics, questionnaire results, and sleep study results between: Group 1, patients completing all three questionnaires and a portable sleep test; Group 2, patients completing both a portable sleep test and outpatient polysomnography; and Group 3, patients completing the STOP-BANG questionnaire and a portable sleep test. All numbers displayed are %, unless otherwise specified.

Clinical Characteristics	Patient Groups		
	Group 1 (*n* = 33)	Group 2 (*n* = 28)	Group 3 (*n* = 49)
Age, years, mean ± SD	57.9 ± 9.9	59.0 ± 11.8	58.4 ± 11.4
Male	59.4	50.0	61.2
Race, white	17.2	22.2	17.0
Race, Black	62.1	59.3	66.0
Ethnicity, Hispanic	13.8	14.8	12.8
Ethnicity, Asian	6.9	3.7	4.3
Body mass index, kg/m^2^ ± SD	30.2 ± 6.8	30.0 ± 6.0	28.7 ± 5.8
Neck circumference, cm ± SD *	38.5 ± 4.3	37.8 ± 5.8	38 ± 5.1
Hypertension	93.9	77.8	85.4
Diabetes mellitus	51.5	51.9	45.8
Ischemic stroke	78.8	74.1	93.8
Intracerebral hemorrhage	3.0	3.7	0
Transient ischemic attack	15.2	18.5	4.2
Subarachnoid hemorrhage	3.0	3.7	2.1
Previous diagnosis of SDB	0	0	2.0
Questionnaires	ESS/BQ/SB	n/a	SB
Epworth Sleepiness Scale ≥10	36.4	-	-
Berlin Questionnaire ≥2	60.6	-	-
STOP-BANG ≥3	81.8	-	77.6
Sleep study results, *n* (%)	PST	PSG	PST
Mild or greater SDB (AHI > 5)	25/33 (75.8)	26/28 (92.9)	37/49 (75.5)
Moderate or greater SDB (AHI > 15)	14/33 (42.4)	16/28 (57.1)	21/49 (42.9)
OSA only, no CSA	17/25 (68.0)	17/26 (65.4)	21/37 (56.8)
CSA only, no OSA	0	0	1/37 (2.7)
Both OSA and CSA	8/25 (32.0)	9/26 (34.6)	15/37 (40.5)

Abbreviations: AHI, apnea hypopnea index; BQ, Berlin Questionnaire; CSA, central sleep apnea; ESS, Epworth Sleepiness Scale; OSA, obstructive sleep apnea; SB, STOP-BANG Questionnaire; SDB, sleep disordered breathing; PST, portable sleep testing; PSG, polysomnography; SD, standard deviation. * Neck circumference data is missing in 20 out of 68 observations.

**Table 2 jcm-10-03568-t002:** Questionnaire Test Parameters for Administered Questionnaires. Comparison of the sensitivity and specificity of BQ, ESS, and SB with 95% CI for the outcome of at least mild sleep disordered breathing (AHI ≥5).

Parameter	Administered Questionnaires		
	BQ (*n* = 33)	ESS (*n* = 33)	SB (*n* = 49)
Sensitivity (95% CI)	0.68 (0.46,0.85)	0.36 (0.18,0.57)	0.81 (0.65,0.92)
Specificity (95% CI)	0.62 (0.24,0.91)	0.62 (0.24,0.91)	0.33 (0.10,0.65)

Abbreviations: AHI, apnea hypopnea index; BQ, Berlin Questionnaire; CI, confidence interval; ESS, Epworth Sleepiness Scale Questionnaire; SB, STOP-BANG Questionnaire.

**Table 3 jcm-10-03568-t003:** Questionnaire Test Parameters for the STOP Questionnaire. The STOP Questionnaire is derived from STOP-BANG administered in Group 3, for secondary analysis. Comparison of the sensitivity and specificity of the STOP Questionnaire using a positive response for one question (STOP ≥1), two questions (STOP ≥2), and three questions (STOP ≥3) as cutoffs, as well as 95% CI, for the outcome of at least mild sleep disordered breathing (AHI ≥5).

Parameter	STOP Questionnaire		
	STOP ≥ 1 (*n* = 49)	STOP ≥ 2 (*n* = 49)	STOP ≥ 3 (*n* = 49)
Sensitivity (95% CI)	0.95 (0.82, 0.99)	0.68 (0.50, 0.82)	0.30 (0.16, 0.47)
Specificity (95% CI)	0.08 (0.00, 0.38)	0.50 (0.21, 0.79)	0.83 (0.52, 0.98)

Abbreviations: AHI, apnea hypopnea index; CI, confidence interval; STOP, STOP Questionnaire.

## Data Availability

Data were not reported.

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
