# Peer review of "Questionnaire and Portable Sleep Test Screening of Sleep Disordered Breathing in Acute Stroke and TIA"

_jcm, 2021, doi:10.3390/jcm10163568_

Round 1

Reviewer 1 Report

Thanks for the opportunity to take another look at this interesting paper, the authors have made a number of important changes.

Overall, the discussion around the lower AHI cutoff has improved a little.  I think most clinicians caring for patients with sleep disordered breathing recognise that the putative benefits of CPAP (or other) treatments simply don't exist at lower AHI thresholds and while I accept the authors adjustments I think that the lower cutoff limits the clinical applicability and impact of this retrospective study. 

I would also like to point out that the authors have mistakenly/erroneously conducted further analysis using AHI 12/hr but if you look at the SAVE trial the SAVE trial authors use a cutoff or 12/hr in ODI4%, this is different to the AHI - I would encourage the authors to conduct this analysis again using ODI4% cutoff of 12 and adjust the manuscript to remove this error- especially given that they have referenced the SAVE trial in this reviewed manuscript.

I am not sure I follow the argument about cutoff points. I don't agree with the argument of using established cutoffs as these were developed by using sensitivity analysis in non-stroke populations. So the question still stands as to why you wouldn't do sensitivity analysis in the stroke population - different cutoffs might deliver a better screening tool in the different population, but the authors don't present this analysis.

Reviewer 2 Report

Thank you for opportunity to review the paper titled Questionnaire and Portable Sleep Test Screening of Sleep Dis- 2 ordered Breathing in Acute Stroke . The paper is well written , flow is good, the statistical methods are appropriate.

However, there are numerous major flaws.

1) No inclusion criteria

2) The group characteristic is very poor. No comorbidity data, i.e. hypertension,which is common in stroke patients. The hypertension might influence the ESS score. The sleepiness was estimated, but data were not presented.

3) The sample is small, the group are heterogenous. Only thirty-seven patients completed all three questionnaires . It is very modest group as in questionnaire study.

4) The describsion of PSG and PST is completly missed.What criteria was used?What sensors? Numer of electrodes? etc

5) In the group 1 and 3 some patients underwent polisomnography , the others home test. It is a serious methodological flaw, because these tests are not comparable.

6) The number of PSG patient and PST from group1 patient is missed

7) How many patients completed ESS i BQ? Information is missed

8) the table1is not legible. Is there prevalence of OSA and CA? If so, the prevalence of CA in group 1 and 2 was 0! It is is very strange in stroke patients but was no discussed

9) The specifity of SQ was very low. The authors overintepreted the results.

10) The results of PSG and PST are missed. The auhors have not showed most of results.

Round 2

Reviewer 2 Report

I reviewed the manuscript titled “ Questionnaire and Portable Sleep Test Screening of Sleep Dis-ordered Breathing in Acute Stroke and TIA”.

The manuscript has been improved as recommended by the reviewer.

Minor points:

  • 105 citiation is missed. Unusual criteria of OSA were used.
  • Figure 2 In OSA authors recommend CPAP auto- tritation, however this method is not  recommended in mild sleep apnea, which is common after stroke. Therefore, surgery  or MADs should also considered.
  • Brief summary is needed. What is a novel result?

Author Response

This manuscript is a resubmission of an earlier submission. The following is a list of the peer review reports and author responses from that submission.

Round 1

Reviewer 1 Report

Thank-you for the opportunity to review this interesting paper. Notwithstanding the inherent limitations of a retrospective analysis I feel that the paper may be improved but significant re-analysis is required.

  • The authors used an AHI cutoff of 5 events/hr. However, an AHI of 5.1/hr is not associated with increased risk of stroke.  What is the justification of using the 5/hr cutoff?  Perhaps the authors could consider an AHI cutoff of 12/hr (using apnealink) which was used in the SAVE trial (McEvoy NEJM 2016) as a measure of moderate/severe OSA (same trial showed subgroup mortality benefit of CPAP so identifying this particular group is more pertinent than the SDB group identified in the current paper)
  • In the methods I think it would be relevant to know at what time during/after stroke admission the sleep studies were performed
  • Additional analyses using alternative questionnaire cutoffs and combining questionnaire answers/results could improve the overall sensitivity. This analysis should be performed and included eg. ROC analysis.
  • The overall aim here is to pilot a screening tool and the current sensitivity of the STOP isn't adequate to act as a screening tool - I think although STOP had the highest sensitivity the authors should temper their language in the discussion and acknowledge that their study is retrospective and that the sensitivity of STOP in this population isn't great.
  • We need to know what proportion of stroke patients are unable to answer the questionnaires due to communication etc issues.  This should be easy to figure out in a retrospective analysis and will help the reader to assess the utility/applicability of the questionnaire approach overall.

Reviewer 2 Report

The manuscript entitled „Questionnaire and portable sleep test screening of sleep disordered breathing in acute stroke” assesses the use of screening questionnaires and portable sleep testing for patients with acute stroke or transient ischemic attack in OSA diagnosis.

Description of statistical analysis needs to be expanded. Data should be first investigated towards distribution. Then, adequate statistical tests should be used to compare groups to see if there are differences between them. Furthermore, no method is stated for assessing sensitivity and specificity of the these (ROC curves?), what were the cutoff used for given sensitivity and specificity provided in the manuscript. Was Youden Index used? Possibly different cutoff points would provide a better sensitivity and specificity.

Considering the size of the group, particularly which underwent all examinations, the study should be considered as pilot study.

The number of assessed questionnaires is quite limited. NoSAS questionnaire should be included in the study, possibly using the data from medical history.

Comparison of obtained data should be disused in context of general population. Otherwise, the article does not provide a perspective of its findings.

It would be interesting to provide information regarding comorbidities of study groups and asses their additional value in diagnosing OSA (fe. DM and HT).

Reference 5 has unnecessary bracket at the beginning.